# UniSeMi: Toward Unified Semi-supervised Medical Image Segmentation

## Abstract

Semi-supervised learning (SSL) for medical image segmentation is put forward to mitigate the scarcity of annotation by leveraging unlabeled data. Recently proposed SSL works focus on designing task-specific models that process different tasks separately. This results in marginal improvement due to inadequate supervision from scarce labels of each single task. To address this, we advocate learning a **Uni**fied **Se**mi-supervised segmentation model for **M**edical **i**maging (UniSeMi) by augmenting the label space, in which all pertinent task data are leveraged simultaneously. Specifically, UniSeMi can complete various missions using one single model with a task-prompted dynamic head. Beyond that, UniSeMi can learn from unlabeled data without requiring associated task information, $i.e.$, which task the unlabeled data belong to remains unknown. To achieve this, we first synthesize an additional task by utilizing labeled data from pertinent tasks, and the synthetic task aims to instruct UniSeMi to be aware of all task semantics. In the context of unlabeled data learning, the aggregated prediction prompted by pertinent tasks is constrained to be consistent with the prediction prompted by the synthetic task, thus task information is not desired. We evaluate UniSeMi on four public medical benchmarks, experiments show UniSeMi outperforms the second-best SSL method by 2.69% and 8.92% according to the averaged Dice and HD score, respectively. Code will be released.

## 1 Introduction

Semi-supervised medical image segmentation is a long-term discussed topic in medical image analysis (Shen et al., 2017; Cheplygina et al., 2019; Jiao et al., 2022; Zeng et al., 2023a). To alleviate the scarcity of data annotation, semi-supervised learning (SSL) first learns from limited labels and then generalizes to abundant unlabeled data. Generally, there are two popular SSL paradigms, called pseudo-labeling and consistency regularization. The first paradigm aims to find out trustworthy pseudo-labels for re-training, $e.g.$, setting an adaptive threshold in terms of the learning process to filter out unreliable predictions (Zhang et al., 2022). And the second focuses on making consistent predictions with smoothness assumption (Van Engelen & Hoos, 2020), which forces the model to make invariant results for the same input but under different augmentations (Sohn et al., 2020; Miyato et al., 2018). Nevertheless, these works are restricted to a specific task where labeled and unlabeled data share a single label space, which results in (1) poor generalization ability when facing out-of-distribution data, and (2) limited performance triggered by inconsistent distribution learned on labeled and unlabeled data. These issues primarily stem from insufficient supervision of each single task, which is especially common in clinical.

Recent investigations into universal models have revealed encouraging results for a variety of tasks, spanning both the computer vision (Kirillov et al., 2023; Xue et al., 2023) and medical imaging communities (Liu et al., 2023; Ye et al., 2023; Xie et al., 2022). Universal models are typically trained using two distinct methods, leveraging data from diverse domains and modalities. The first approach involves pre-training the model using task-agnostic unlabeled data through self-supervised learning, followed by fine-tuning on task-specific data for individual downstream tasks (Wang et al., 2023; Xie et al., 2022; Zhou et al., 2022). The second method trains a model jointly using multiple task-specific data in a supervised fashion (Chen et al., 2023b; Ye et al., 2023; Zhang et al., 2021). Both approaches have shown superior performance than single models, further emphasizing the importance of integrating data for enhanced representation learning. Based on our observations,

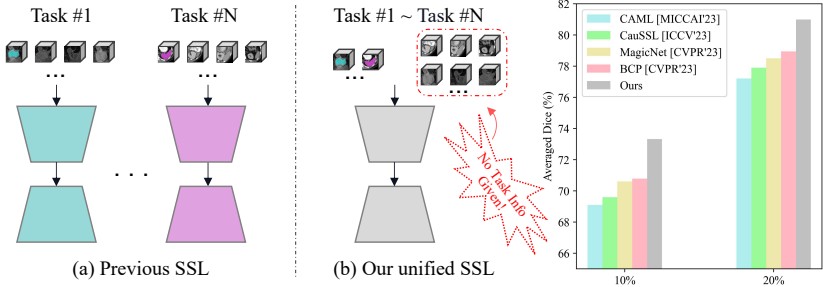

Figure 1: A brief view of previous SSL methods (a) and our method (b). Compared to previous methods where a separate SSL model was designed for each task, we advocate for using a single, universal model to tackle all these SSL tasks. This unified model does not necessitate specific task information when learning from unlabeled data. The figure on the right displays the average Dice score across four SSL segmentation tasks, using label percentages of 10% and 20%. It is evident that our method surpasses other competitors by a significant margin.

there is a pressing need to develop a unified SSL framework. Previous SSL research has largely centered on addressing each task in isolation, neglecting the potential of jointly harnessing data from diverse sources in a semi-supervised context. In contrast to individual SSL models, a unified framework presents multiple advantages. It offers not just a versatile, all-encompassing solution but also improved performance stemming from the availability of more labeled and unlabeled data.

Driven by the analysis, in this paper, we propose a **Uni**fied **S**emi-supervised **S**egmentation Model (UniSeMi) for medical imaging. UniSeMi seamlessly establishes an enhanced label space by amalgamating pertinent task labels, when confronted with multiple SSL tasks. Specifically, UniSeMi incorporates a versatile dynamic segmentation head that learns to segment task-of-interest objects, *i.e.*, organs and tumors, with task-specific prompts. With these prompts as a guide, UniSeMi can flexibly generate parameters for the dynamic head, enabling it to adapt and effectively execute the ongoing task. Furthermore, we position UniSeMi in a more demanding SSL context, in which the task specifics of unlabeled data remain unknown during the training process (see Fig. 1b) To navigate this challenge, UniSeMi employs the cutmix data augmentation technique on labeled data from various pertinent tasks, producing synthetic data. This data spans a diverse range of foreground targets within the expanded label space. Therefore, the synthetic task with synthetic data can direct UniSeMi to recognize and segment potential foreground regions. When processing unlabeled data, UniSeMi strives to align predictions generated from specific tasks with those derived from the synthetic task, ensuring consistency across mixed unlabeled datasets. Hence, task-agnostic unlabeled data learning is achieved by devising a synthetic task. Based on the aforementioned strategies, UniSeMi is able to work in a more demanding SSL context and surpasses the second-best method by 2.69% and 8.92% on the average Dice and HD scores, when using 10% labels.

Our contributions are in three aspects: (1) We propose UniSeMi, a universal semi-supervised framework tailored for medical image segmentation. Leveraging a dynamic head, this framework is adept at concurrently optimizing multiple semi-supervised tasks, achieving this with both efficiency and precision; (2) We devise a "synthetic task" that amalgamates labeled data from several pertinent tasks. This design facilitates learning of unified foreground segmentation and subsequently uses this segmentation ability as a constraint to mine unlabeled data from multiple sources; and (3) Extensive experiments on four public datasets validate the superiority of UniSeMi, which presents remarkable improvements when compared to the prior methods.

## 2 RELATED WORK

### 2.1 SEMI-SUPERVISED LEARNING

Semi-supervised learning (SSL) (Chen et al., 2022b; Mey & Loog, 2022; Cheplygina et al., 2019) is emerged to mitigate the issue of tedious data annotation, by learning from unlabeled data with scarce labeled data. Many efforts are made to explore how to excavate information from unlabeled data adequately. For instance, (Rizve et al., 2020) reduced unreliable pseudo-labels by calibrating mod-

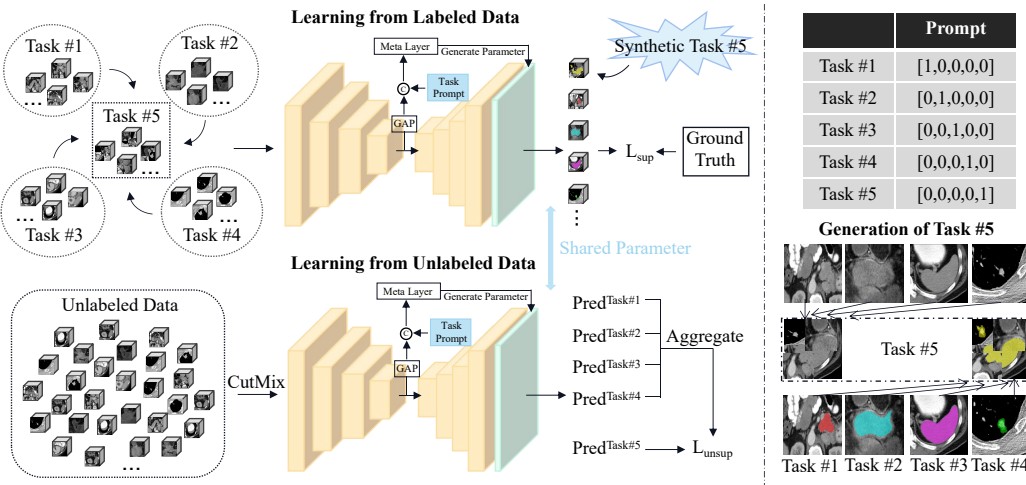

Figure 2: Illustration of UniSeMi. UniSeMi learns from all tasks simultaneously and doesn't require task information in context of unlabeled data.

els with uncertainty. (Zeng et al., 2023b) investigated the probability distribution of pseudo-labeled data, and further proposed a selection standard from the perspective of loss distribution. (Chen et al., 2022a; Wang et al., 2022) tried to address the quantity-quality trade-off issue with adaptive threshold. Although these works present improved results compared to the prior, the gap between the supervised baseline still can not be ignored. The main reason for the phenomenon is perhaps that the learned representation is insufficient since there are usually limited labels in each dataset. To address the mentioned issue, we advocate learning a unified SSL model with an integrated dataset, where all procurable data from various tasks are leveraged at the same time.

## 2.2 REPRESENTATION LEARNING WITH INTEGRATED DATA

To improve the model performance and representation ability, some works propose to learn a unified model that can complete multiple tasks simultaneously, rather than training task-specific model separately (Kim et al., 2022; Lee et al., 2023). For example, (Zhang et al., 2021) collected an abdominal dataset from seven partially labeled datasets for model training, and presented better-averaged results than training on every single dataset. (Liu et al., 2023) further advanced this idea by introducing CLIP embedding (Radford et al., 2021) to help the model capture anatomical relations between different tumors and organs. (Ye et al., 2023) leveraged different modalities including CT, MRI and PET, whose performance surpassed by those models trained with a single modality. These studies underscore the importance of robust data engines, emphasizing the need to leverage as much accessible data as possible. While the majority of these investigations focus on either fully supervised learning (Ye et al., 2023; Liu et al., 2023) or self-supervised learning (Xie et al., 2022), we put forth a unified SSL model designed to harness the strengths of both labeled and unlabeled data.

## 3 METHOD

Fig. 2 presents the workflow of UniSeMi, which is developed to complete kinds of segmentation tasks using one single model under SSL setting. Compared to existing SSL works, UniSeMi can learn from unlabeled data without knowing associated task information. Below, Section 3.1 describes the flexibility of UniSeMi for its usage of dynamic kernel generation. Section 3.2 and Section 3.3 present supervised and unsupervised learning on labeled and unlabeled data, respectively.

## 3.1 DYNAMIC CONVOLUTION WITH TASK PROMPT

As pointed out by (Zhang et al., 2021; Lei et al., 2022), the fixed kernel parameters optimized on one task are not optimal for the others, and it is always a challenge to use a single model to complete the segmentation of multiple organs and tumors. One intuitive way to address this issue is using a multi-head model, but it suffers from severe computational overhead with the increase of on-coming

tasks, thus is not suitable as a universal model. In this work, we adopt dynamic filter generation to initialize the segmentation head, which can adaptively process different tasks with task-specific prompts while the computational cost remains unchanged. The parameter generation is defined as:

$$w_k = \psi(GAP(Embedding), [Prompt_{\#k}]; \theta_\psi), \tag{1}$$

$$\mathcal{P}_k = SoftMax(f_D(Embedding) * w_k), \tag{2}$$

where $\psi_\theta$ is one convolutional layer with parameter $\theta$, which is employed to dynamically generate parameters $w_k$ for the current Task#k with $[Prompt_{\#k}]$. Here one-hot encoding is used as a prompt, which is concatenated with global averaged feature embedding before feeding into $\psi_\theta$. $\mathcal{P}_k$ is the prediction for Task#k, $f_D$ is the decoder and symbol $*$ represents convolution. With $[Prompt_{\#k}]$, UniSeMi can accurately perceive the ongoing task and flexibly adapt kernels to fit it.

### 3.2 Labeled Data Learning with Additionally Synthesized Task

As the bottom-right of Fig. 2 shows, we construct a synthetic task (Task#5) based on labeled data from pertinent tasks (Task#1 $\sim$ Task#4). Task#5 is built to help UniSeMi achieve task-agnostic unlabeled data learning, as well as guiding UniSeMi to segment all foreground regions from mixed data. The data generation of Task#5 is formulated as:

$$\mathcal{X}_{syn}^l = \mathcal{X}_i^l \odot \mathcal{M} + \mathcal{X}_j^l \odot (1 - \mathcal{M}), \tag{3}$$

$$\mathcal{Y}_{syn}^l = \mathcal{Y}_i^l \odot \mathcal{M} + \mathcal{Y}_j^l \odot (1 - \mathcal{M}), \tag{4}$$

where $\mathcal{X}_{syn}^l$ and $\mathcal{Y}_{syn}^l$ are synthetic images and labels for Task#5. $\mathcal{M}$ is a mask with 30% $\sim$ 70% random masked regions. Symbol $\odot$ is element-wise multiplication. Note that $\mathcal{Y}_i^l$ and $\mathcal{Y}_j^l$ are binary masks, $\mathcal{X}_i^l$ and $\mathcal{X}_j^l$ are images from the $i$-th and $j$-th task, thus $\mathcal{X}_{syn}^l$ can be regarded as mixed data that contain various targets and background. For labeled data learning (containing Task#5), Dice loss and cross-entropy loss are leveraged, defined as:

$$\mathcal{L}_{sup} = \mathcal{L}_{Dice}(\mathcal{F}(\mathcal{X}_k^l, [Prompt_{\#k}]; \Theta), \mathcal{Y}_k^l) + \mathcal{L}_{ce}(\mathcal{F}(\mathcal{X}_k^l, [Prompt_{\#k}]; \Theta), \mathcal{Y}_k^l), \tag{5}$$

where $\mathcal{L}_{sup}$ is the supervised loss on labeled data. For simplicity, we use $\mathcal{F}(\cdot; \Theta)$ to define the unified model with parameter $\Theta$, which contains operations in Eq.1 and Eq.2. Based on the above procedure, UniSeMi ought to have a semantic perception of all tasks if prompted by Task#5 (Fig. 4 presents a clear view).

### 3.3 Unlabeled Data Learning with Self-consistency

Thanks to the design of semantic-aware Task#5, the unified SSL model learns from unlabeled data without knowing the associated task information. Below we describe how this is achieved. Firstly, CutMix (Yun et al., 2019) is conducted on all unlabeled data, making the input contain objects of different tasks. Then the prediction with **Task#5 Prompt** is forced to be consistent with the aggregated prediction using **Task#1 Prompt** $\sim$ **Task#4 Prompt**. The aggregated prediction can be regarded as a combination of pseudo-masks for each task, and the prediction prompted by Task#5 can be considered as a direct pseudo-mask for all tasks. Therefore, the two predictions should be identical. We call this operation self-consistency since no extra decoder or teacher model is required for supervision. The entire process can be written as:

$$\mathcal{X}_{syn}^u = \mathcal{X}_1^u \odot \mathcal{M} + \mathcal{X}_2^u \odot (1 - \mathcal{M}), \tag{6}$$

$$\mathcal{P}_{agg} = \max_{k \in (1,4)} (\mathcal{F}(\mathcal{X}_{syn}^u, [Prompt_{\#k}]; \Theta)), \tag{7}$$

where $\mathcal{X}_{syn}^u$ are mixed unlabeled data, $\mathcal{X}_1^u$ and $\mathcal{X}_2^u$ are randomly selected unlabeled data. Element-wise maximization is performed to aggregate predictions prompted by Task#1 $\sim$ Task#4, and $\mathcal{P}_{agg}$ is the final aggregated prediction. The overall loss $\mathcal{L}$ and unsupervised loss $\mathcal{L}_{unsup}$ are calculated as:

Table 1: Performance comparison with other seven state-of-the-art methods on the pancreas dataset, in the scenario of leveraging 10% and 20% labeled data. The best and second best results are shown in red and blue, respectively. Results using unlabeled task information are highlighted in gray .

| Method | Pancreas (10%/6 labeled data) | | | | Pancreas (20%/12 labeled data) | | | |
|---|---|---|---|---|---|---|---|---|
| | Dice ↑ | Jaccard ↑ | ASD ↓ | 95HD ↓ | Dice ↑ | Jaccard ↑ | ASD ↓ | 95HD ↓ |
| VNet | 55.60 | 41.74 | 18.63 | 45.33 | 72.38 | 58.26 | 5.89 | 19.35 |
| UA-MT (MICCAI'19) | 66.34 | 53.21 | 4.57 | 17.21 | 76.10 | 62.62 | 2.43 | 10.84 |
| DTC (AAAI'21) | 69.21 | 54.06 | 5.95 | 17.21 | 78.27 | 64.75 | 2.25 | 8.36 |
| ASE-Net (TMI'22) | 71.54 | 56.82 | 5.73 | 16.33 | 79.03 | 66.57 | 2.30 | 8.62 |
| CAML (MICCAI'23) | 71.21 | 56.32 | 5.92 | 16.89 | 79.81 | 67.35 | 2.27 | 8.22 |
| BCP (CVPR'23) | 73.83 | 59.24 | 3.72 | 12.71 | 82.91 | 70.97 | 2.25 | 6.43 |
| CauSSL (ICCV'23) | 72.34 | 57.43 | 3.13 | 13.49 | 80.63 | 67.84 | 2.78 | 8.76 |
| MagicNet (CVPR'23) | 75.01 | 62.04 | 3.97 | 13.71 | 81.25 | 68.81 | 2.83 | 8.50 |
| UniSeMi (Ours) | 78.08 | 64.82 | 2.33 | 8.05 | 83.27 | 71.68 | 1.40 | 5.33 |
| UniSeMi w/ Task Info | 78.62 | 64.91 | 2.28 | 7.99 | 83.55 | 71.93 | 1.35 | 5.02 |

Table 2: Performance comparison with other seven state-of-the-art methods on the left atrium dataset, in the scenario of leveraging 10% and 20% labeled data. The best and second best results are shown in red and blue, respectively.

| Method | Left Atrium (10%/8 labeled data) | | | | Left Atrium (20%/16 labeled data) | | | |
|---|---|---|---|---|---|---|---|---|
| | Dice ↑ | Jaccard ↑ | ASD ↓ | 95HD ↓ | Dice ↑ | Jaccard ↑ | ASD ↓ | 95HD ↓ |
| VNet | 82.74 | 71.72 | 3.26 | 13.35 | 84.89 | 77.32 | 2.97 | 11.60 |
| UA-MT (MICCAI'19) | 86.28 | 76.11 | 4.63 | 18.71 | 88.74 | 79.94 | 2.32 | 8.39 |
| DTC (AAAI'21) | 87.51 | 78.17 | 2.36 | 8.23 | 89.42 | 80.89 | 2.10 | 7.32 |
| ASE-Net (TMI'22) | 87.83 | 78.45 | 2.17 | 9.86 | 90.29 | 82.76 | 1.64 | 7.18 |
| CAML (MICCAI'23) | 89.62 | 81.28 | 2.02 | 8.76 | 90.78 | 83.19 | 1.68 | 6.11 |
| BCP (CVPR'23) | 89.62 | 81.31 | 1.76 | 6.81 | 91.25 | 83.85 | 1.47 | 5.96 |
| CauSSL (ICCV'23) | 88.37 | 79.50 | 2.74 | 9.24 | 90.46 | 82.37 | 1.96 | 6.62 |
| MagicNet (CVPR'23) | 88.65 | 79.89 | 3.01 | 9.78 | 90.17 | 82.24 | 2.14 | 7.83 |
| UniSeMi (Ours) | 89.01 | 80.52 | 2.57 | 9.03 | 90.89 | 83.48 | 1.72 | 5.38 |
| UniSeMi w/ Task Info | 89.83 | 81.27 | 2.38 | 8.62 | 91.29 | 84.08 | 1.86 | 5.62 |

$$\mathcal{L} = \mathcal{L}_{sup} + \mathcal{L}_{unsup}, \tag{8}$$

$$\mathcal{L}_{unsup} = \mathcal{L}_{Dice}(\mathcal{P}_{agg}, \mathcal{F}(\mathcal{X}^u_{syn}, [Prompt_{\#5}]; \Theta)). \tag{9}$$

## 4 EXPERIMENTS

### 4.1 SETUP

**Datasets.** We report the model segmentation results on four public datasets, including **Task#1:** NIH-Pancreas (Roth et al., 2015), **Task#2:** Left Atrium (Xiong et al., 2021), **Task#3:** MSD-Spleen (Antonelli et al., 2022) and **Task#4:** MSD-Lung Tumor (Antonelli et al., 2022). Specifically, NIH-Pancreas contains 82 contrast-enhanced abdomen CT scans, which are split into 62/20 scans for training/test. The Left Atrium has 100 gadolinium-enhanced MR images, in which 80/20 images are leveraged for training/test. MSD-Spleen contains 41 CT scans, and 30/11 scans are split for training/test. MSD-Lung Tumor contains 63 CT scans, which are divided into 50/13 scans for training/test. All methods follow the same data split for fair comparisons, with the same pre-processing as (Luo et al., 2021; Bai et al., 2023).

**Implementation Details.** Following previous works (Bai et al., 2023; Luo et al., 2021; Chen et al., 2023a), V-Net (Milletari et al., 2016) was adopted as the baseline model for fair comparisons. Adam optimizer was used with a learning rate of 0.001. The input size and batch size were set to 96×96×96 and 8, respectively. Experiments were implemented by Pytorch (Paszke et al., 2019) with four NVIDIA GeForce RTX 3080 Ti GPUs. Evaluation metrics of Dice, Jaccard, Average Surface Distance (ASD) and 95% Hausdorff Distance (95HD) were reported.

Table 3: Performance comparison with other seven state-of-the-art methods on the spleen dataset, in the scenario of leveraging 10% and 20% labeled data. The best and second best results are shown in red and blue, respectively.

| Method | Spleen (10%/3 labeled data) | | | | Spleen (20%/6 labeled data) | | | |
|---|---|---|---|---|---|---|---|---|
| | Dice ↑ | Jaccard ↑ | ASD ↓ | 95HD ↓ | Dice ↑ | Jaccard ↑ | ASD ↓ | 95HD ↓ |
| VNet | 75.14 | 65.27 | 15.02 | 43.89 | 79.78 | 72.86 | 11.37 | 30.03 |
| UA-MT (MICCAI'19) | 79.63 | 68.62 | 15.94 | 44.71 | 83.11 | 75.98 | 8.92 | 25.41 |
| DTC (AAAI'21) | 80.27 | 69.00 | 14.53 | 41.56 | 84.59 | 75.91 | 9.75 | 31.77 |
| ASE-Net (TMI'22) | 80.65 | 69.48 | 14.37 | 41.31 | 85.02 | 75.68 | 12.53 | 37.26 |
| CAML (MICCAI'23) | 80.32 | 69.10 | 15.37 | 41.71 | 85.80 | 76.79 | 11.57 | 36.14 |
| BCP (CVPR'23) | 83.12 | 72.85 | 14.42 | 42.11 | 87.02 | 78.58 | 10.48 | 37.08 |
| CauSSL (ICCV'23) | 81.98 | 71.25 | 14.69 | 41.84 | 86.83 | 78.46 | 10.01 | 32.27 |
| MagicNet (CVPR'23) | 83.55 | 73.58 | 13.49 | 41.79 | 88.24 | 80.24 | 8.50 | 23.51 |
| UniSeMi (Ours) | 89.34 | 81.73 | 3.12 | 9.33 | 94.62 | 89.89 | 2.40 | 7.50 |
| UniSeMi w/ Task Info | 90.10 | 82.75 | 3.09 | 9.28 | 94.67 | 89.93 | 2.35 | 7.33 |

Table 4: Performance comparison with other seven state-of-the-art methods on the lung tumor dataset, in the scenario of leveraging 10% and 20% labeled data. The best and second best results are shown in red and blue, respectively.

| Method | Lung Tumor (10%/5 labeled data) | | | | Lung Tumor (20%/10 labeled data) | | | |
|---|---|---|---|---|---|---|---|---|
| | Dice ↑ | Jaccard ↑ | ASD ↓ | 95HD ↓ | Dice ↑ | Jaccard ↑ | ASD ↓ | 95HD ↓ |
| VNet | 31.07 | 21.69 | 14.81 | 24.16 | 36.89 | 26.00 | 12.98 | 23.47 |
| UA-MT (MICCAI'19) | 33.46 | 25.88 | 15.08 | 24.78 | 44.33 | 30.80 | 10.28 | 22.82 |
| DTC (AAAI'21) | 34.97 | 26.82 | 12.88 | 24.34 | 48.46 | 34.49 | 7.70 | 21.22 |
| ASE-Net (TMI'22) | 34.18 | 25.86 | 13.09 | 22.91 | 53.15 | 38.29 | 3.77 | 12.87 |
| CAML (MICCAI'23) | 35.24 | 22.99 | 12.33 | 24.25 | 52.43 | 37.03 | 4.07 | 12.65 |
| BCP (CVPR'23) | 36.60 | 27.69 | 11.71 | 23.86 | 54.63 | 38.13 | 3.62 | 11.77 |
| CauSSL (ICCV'23) | 35.72 | 26.25 | 12.52 | 24.09 | 53.69 | 38.47 | 5.05 | 13.40 |
| MagicNet (CVPR'23) | 35.24 | 22.99 | 12.33 | 24.25 | 54.32 | 38.58 | 3.72 | 13.04 |
| UniSeMi (Ours) | 36.90 | 28.12 | 10.87 | 23.41 | 55.16 | 42.47 | 6.39 | 16.75 |
| UniSeMi w/ Task Info | 37.91 | 29.05 | 9.20 | 22.45 | 56.82 | 44.97 | 6.32 | 14.25 |

## 4.2 COMPARISON WITH EXISTING METHODS

We compare our UniSeMi with seven popular SSL methods, including uncertainty-aware mean-teacher (UA-MT) (Yu et al., 2019), dual-task consistency (DTC) (Luo et al., 2021), adversarial consistency and dynamic convolution (ASE-Net) (Lei et al., 2022), correlation-aware mutual learning (CAML) (Gao et al., 2023), bidirectional copy-paste (Bai et al., 2023), causality-inspired semi-supervised segmentation (CauSSL) (Miao et al., 2023) and cubic volume partition and recovery (Magic-Net) (Chen et al., 2023a).

**Results on Pancreas Dataset.** As shown in Table 1, we can find UniSeMi consistently surpasses competitors on all metrics under different label percentages. For example, UniSeMi brings respectively 3.07% and 5.66% improvements on Dice and HD score, when compared to the second best method MagicNet with 10% labeled data. Beyond that, we can find the performance gains obtained by UniSeMi are larger than the others when leveraging fewer labels. This phenomenon indicates that UniSeMi is more applicable in annotation-efficient scenario.

**Results on Left Atrium Dataset.** As indicated by results in Table 2, UniSeMi achieves the second place and BCP becomes the best one, but it is worth noting that the performance gap between the two methods is marginal, usually less than 0.5% on Dice score. Moreover, we observe that all methods can achieve nearly similar Dice score no matter using 10% or 20% labels, and the performance gap between each other is also negligible. This result is mainly attributed to the larger size of dataset and more slices used in each case, thus the model gains sufficient supervision from labeled data and makes correct predictions on unlabeled data.

**Results on Spleen Dataset.** Table 3 presents the results of spleen segmentation. It is a surprise to see that UniSeMi outperforms other competitive methods by a large margin. For instance, compared to MagicNet, UniSeMi has respectively 32.46% and 16.01% performance gains on HD score with 10% and 20% labeled data. Similarly, UniSeMi surpasses CauSSL by 7.36% on Dice score under 10% label percentage.

Table 5: A case study of the impact of other tasks on one specific task. Here spleen segmentation task (Task #3) is selected as the baseline, as we find UniSeMi presents remarkable improvements on this task when compared to other methods, and the purpose of this experiment is to figure out where the performance gains come from.

| Setting | Spleen (10%/3 labeled data) | | | | Spleen (20%/6 labeled data) | | | |
|---|---|---|---|---|---|---|---|---|
| | Dice ↑ | Jaccard ↑ | ASD ↓ | 95HD ↓ | Dice ↑ | Jaccard ↑ | ASD ↓ | 95HD ↓ |
| Task #3 | 75.14 | 65.27 | 15.02 | 43.89 | 79.78 | 72.86 | 11.37 | 30.03 |
| Task #3 + Task #1 | 85.62 | 75.96 | 4.90 | 17.07 | 90.00 | 82.18 | 5.27 | 15.81 |
| Task #3 + Task #1 + Task #2 | 88.03 | 79.26 | 3.78 | 11.06 | 92.06 | 86.39 | 3.41 | 10.06 |
| Task #3 + Task #1 + Task #2 + Task #4 | 89.34 | 81.73 | 3.12 | 9.33 | 94.62 | 89.89 | 2.40 | 7.50 |

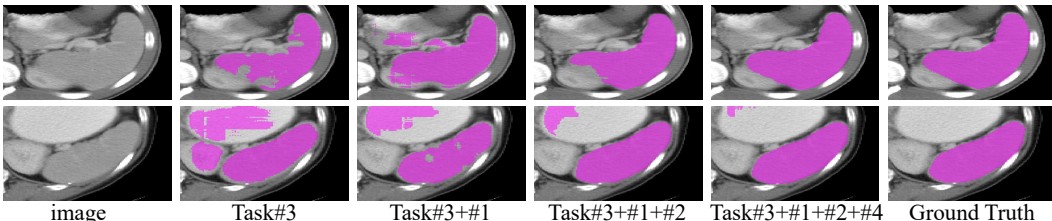

| image | Task#3 | Task#3+#1 | Task#3+#1+#2 | Task#3+#1+#2+#4 | Ground Truth |

Figure 3: Visualization results of pancreas segmentation by incorporating other tasks sequentially. We can find the model tends to produce more accurate segmentation with the increase of integrated tasks, demonstrating the benefits of learning a unified model.

To figure out where the gains come from, we conducted a case study to see the impact of other tasks on spleen segmentation (Task#3). As Table 5 and Fig. 3 show, there are consistent improvements by gradually integrating data of other tasks. In particular, we can find the results increase significantly by singly integrating pancreas segmentation (Task#1), whose performance has already outperformed other competitive methods like MagicNet and BCP. Here we provide three potential reasons for UniSeMi's high performance on the spleen segmentation task: (1) due to extremely limited labels (10% labels are equal to 3 labeled data), competitors fail to generalize the representation learned from labeled data to unlabeled data, and mistakenly predict the background as foreground (see Appendix.D). Therefore, a high HD score can be observed; (2) owing to the same modality, $e.g.$, Task#1 and Task#3, the unified model can learn modality-related knowledge and present better results. (see Fig. 7, the feature embedding of pancreas and spleen are very close in the latent space,

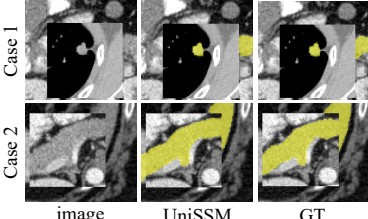

Figure 4: Predictions made by UniSeMi when facing mixed data. **Case 1** is the combination of left atrium and lung tumor. **Case 2** is the combination of pancreas and spleen. We can find UniSeMi can produce accurate masks if prompted by Task#5.

both of them are abdominal organs); and (3) despite the foreground information of each task does not increase, the various mixed background information can guide UniSeMi to learn which region should not be segmented, thereby decreasing the HD score.

**Results on Lung Tumor Dataset.** We also conduct experiments on the lung tumor segmentation task. As Table 4 presents, our proposed UniSeMi achieves first place with 10% labeled data, and surpasses MagicNet by 5.13% of Jaccard score. Under the label percentage of 20%, UniSeMi again produces the highest Dice and Jaccard scores but also shows higher ASD and HD scores. We think this is mainly caused by the misclassified tissues around the tumor and therefore the metrics related to surface distance are increased. Whereas it should be noted that UniSeMi still has 3.89% Jaccard performance gains with 20% label percentage, when compared to MagicNet.

**Summary.** Grounded on the segmentation results on four tasks, we can draw some findings as follows: (1) The proposed UniSeMi presents obvious improvements compared to other SSL methods. For instance, according to the averaged Dice and HD score, UniSeMi respectively surpasses the second-best method BCP by 2.69% and 8.92% with 10% labels, demonstrating the necessity of learning a universal SSL model. (2) For the task with large dataset size, $e.g.$, Task#2, most SSL methods are comparable, including UniSeMi. (3) For the task with a smaller dataset size, $e.g.$, Task#3, the unified model presents remarkable results compared to single models.

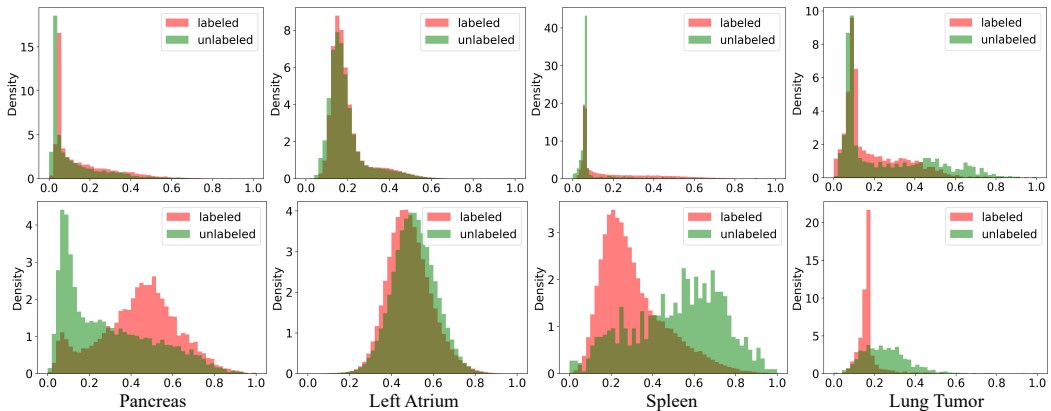

Figure 5: Kernel density estimation of UniSeMi and BCP when training with 10% labels. This experiment is conducted to see the distribution of labeled and unlabeled data. **Top** is the distribution generated by UniSeMi and **Bottom** is BCP (Bai et al., 2023). It is clear to see that UniSeMi aligns the distribution better than BCP.

## 4.3 IN-DEPTH ANALYSIS

**Incorporating unlabeled task information into UniSeMi.** To explore the upper bound of UniSeMi, we report the results when feeding task information of unlabeled data into UniSeMi. The last row of Table 1 ∼ Table 4 with gray background are corresponding results. Given unlabeled task information, UniSeMi can directly generate pseudo-mask based on task-specific data instead of mixed data. From the results, we can find there is an improvement (0.78% gain on the averaged Dice score with 10% labels) but not quite large, demonstrating UniSeMi has the ability to produce accurate prediction when facing mixed data. (More description can be found in Appendix.B).

**Copy-past choices of Task#5.** Task#5 is generated based on CutMix, below we discuss the form of cropped regions. As Table 7 shows, three types of masks are included, containing random mask, slice-based mask and zero-centered mask (see Fig.6).

The first one means cropping several random cubic patches with a shape of 6×6×6. The second one stands for generating slice-wise masks, which can be regarded as slice exchange between different unlabeled data. The last one represents cropping one larger patch. As indicated by the results, the random mask has the worst performance, as the randomly generated cubic patches do not contain coherent foreground informa-

Figure 6: Copy-past choices.

tion, and can be easily confused with the background of other tasks. Alternatively, the slice-based mask shows better results than the random mask, since the integrity of the foreground is reserved. By comparison, the zero-centered mask performs best, as the foreground and background of different tasks have interacted while maintaining the coherence of targets. Therefore, CutMix with the zero-centered mask is leveraged during the construction of Task#5.

Table 6: Adapting BCP and CauSSL into universal models. However, severe performance degradation can be seen when comparing Uni-BCP to BCP (Uni-CauSSL to CauSSL).

| Method | Pancreas (10%/6 labeled data) | | | | Left Atrium (10%/8 labeled data) | | | |
|---|---|---|---|---|---|---|---|---|
| | Dice ↑ | Jaccard ↑ | ASD ↓ | 95HD ↓ | Dice ↑ | Jaccard ↑ | ASD ↓ | 95HD ↓ |
| Uni-BCP | 68.59 | 53.73 | 7.33 | 20.62 | 85.73 | 75.06 | 10.17 | 30.33 |
| Uni-CauSSL | 65.35 | 49.09 | 6.16 | 20.89 | 83.40 | 72.43 | 8.84 | 34.94 |
| UniSeMi (Ours) | 78.08 | 64.82 | 2.33 | 8.05 | 89.01 | 80.52 | 2.57 | 9.03 |
| Method | Spleen (10%/3 labeled data) | | | | Lung Tumor (10%/5 labeled data) | | | |
| | Dice ↑ | Jaccard ↑ | ASD ↓ | 95HD ↓ | Dice ↑ | Jaccard ↑ | ASD ↓ | 95HD ↓ |
| Uni-BCP | 74.80 | 58.89 | 17.11 | 54.06 | 31.01 | 21.32 | 11.35 | 24.36 |
| Uni-CauSSL | 73.06 | 57.85 | 18.28 | 55.51 | 25.38 | 20.20 | 15.06 | 28.72 |
| UniSeMi (Ours) | 89.34 | 81.73 | 3.12 | 9.33 | 36.90 | 28.12 | 10.87 | 23.41 |

Figure 7: t-SNE visualization of feature embedding for four tasks. The implemented Uni-CauSSL, Uni-BCP and our proposed UniSeMi are compared.

**Learned distribution on labeled and unlabeled data.** Distribution mismatch between labeled and unlabeled data is a commonly encountered issue in SSL, which is mainly caused by unbalanced/partial distribution learned from labeled data (Zeng et al., 2023b; Chen et al., 2023a). Fig. 5 presents the kernel density estimation of UniSeMi and BCP when training with 10% label percentage. We can find that: (1) for Task#2 (left atrium segmentation) with large data scale, both UniSeMi and BCP show much-aligned distribution, which is mainly attributed to ample representation learned from labeled data, thus models can successfully generalize to unlabeled data and show comparable performance; (2) for the rest tasks, severe inconsistent distribution is observed for single model BCP, whereas UniSeMi significantly aligns the learned distribution. This demonstrates that properly learning a universal model is beneficial to unlabeled data mining, since the mismatch issue between labeled and unlabeled data is largely alleviated.

**Can other SSL methods process all tasks simultaneously?** To answer this question, we revise CauSSL and BCP into the form of the universal model. There are two changes compared to their previous versions. (1) The input data cover four tasks and are randomly fed into the model with the associated task id. (2) Changing the number of output channels to match the number of tasks, which is different from UniSeMi as UniSeMi has a dynamic task-prompted head with two output channels. As Table 6 shows, the results produced by Uni-BCP and Uni-CauSSL are far inferior to UniSeMi, and compared to their original single model version, significant performance degradation can be observed. For instance, according to the averaged Dice score with 10% labeled data, BCP vs Uni-BCP (70.79% vs 65.03%), 5.76% drop can be found. And CauSSL vs Uni-CauSSL (69.60% vs 61.80%), 7.80% degradation is discovered. This phenomenon is mainly triggered by chaotic representation learned from all task data, and also indicates that naively learning from all tasks simultaneously is not effective and even harmful to the single task. Moreover, we plot the t-SNE visualization of feature embedding to have a clear view. As Fig. 7 exhibits, UniSeMi presents a distinguishable decision boundary while others show mixed and dispersed embedding. This demonstrates that task-specific prompts are essential, especially in learning a universal model, as such prior can guide the model to have a clear understanding of the ongoing task, thus showing discernible and concentrated representation for each single task.

Table 7: Design choices of copy-past for the generation of Task#5. The averaged Dice and 95HD scores on four tasks are reported under label percentage of 10%.

| Strategy | Dice ↑ | 95HD ↓ |
|---|---|---|
| Random mask | 70.21 | 15.32 |
| Slice-based mask | 72.16 | 13.88 |
| Zero-centered mask | **73.33** | **12.46** |

## 5  CONCLUSION

In this paper, we propose a semi-supervised segmentation model UniSeMi for medical imaging. Compared to existing SSL works, UniSeMi can flexibly process kinds of tasks using one single model, and can learn from unlabeled data without requiring associated task information. Benefiting from the task-prompted dynamic head, UniSeMi successfully learns dividable feature embedding for different tasks, while the representation learned by other implemented universal models (Uni-BCP, Uni-CauSSL) is in chaos. Extensive experiments validate the effectiveness of UniSeMi especially for tasks with few labels. **Limitations:** We can find UniSeMi also produces mixed embedding (see Fig. 7) and the model performance sometimes falls behind single models. We think this is attributed to conflicts among data of tasks. Despite standardized normalization being conducted, a de-biased strategy for data processing may be helpful to further mitigate the contradiction.

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
