## A    WHY NOT USING MIXUP TO SYNTHETIZE TASK#5?

Let's first review the role of the synthetic Task#5. It is constructed to pave the way for task-agnostic unlabeled learning. Therefore, the warm-upped model on labeled data ought to be aware of all task semantics since the model doesn't know the task information when facing unlabeled data. Thus, the data belong to Task#5 should contain information of all existing tasks.

Empirically, Mixup, CutMix and their variants are well-acknowledged copy-past strategies to synthesize new training data by mixing existing data. In our work, Mixup-based schemes are not suitable for the following reasons: (1) Mixup works in the way of mixing whole images with different weights, so there are two types of voxels that contain task information, one is the mixed foreground-foreground voxel and the other is the mixed foreground-background voxel, whereas the latter can not be regarded as a pure target. (2) Therefore, models trained using Mixup-based data can not clearly realize the task semantics. Whereas CutMix doesn't have such issues.

## B    DIFFERENCE BETWEEN UNISEMI AND UNISEMI W/TASK INFO

As Fig. 1 (a) shows, UniSeMi is designed to learn from unlabeled data without knowing associated task information. And in experiments, we also present the results of UniSeMi w/ task info, whose workflow is exhibited by Fig. 1 (b). Specifically, UniSeMi w/ task_info directly makes prediction on the source image with task-specific prompt, whereas UniSeMi w/o task_info should first make predictions on the mixed data with all pertinent task prompts and then aggregate them.

## C    SEGMENTATION RESULTS PROMPTED BY PERTINENT TASKS

Fig. 2 presents the segmentation results, which are produced using task-specific prompt. We can find that under the control of task-specific prompt, UniSeMi can accomplish associated segmentation task and is hardly affected by other tasks. (But there are also some exceptions. For example, in case 1, when confronted with the mixed unlabeled data of left atrium and spleen, predictions prompted by pancreas segmentation and lung tumor segmentation are slightly highlighted on the region of spleen. We think think is mainly attributed to the correlation between tasks, but it doesn't affect the final aggregated results.)

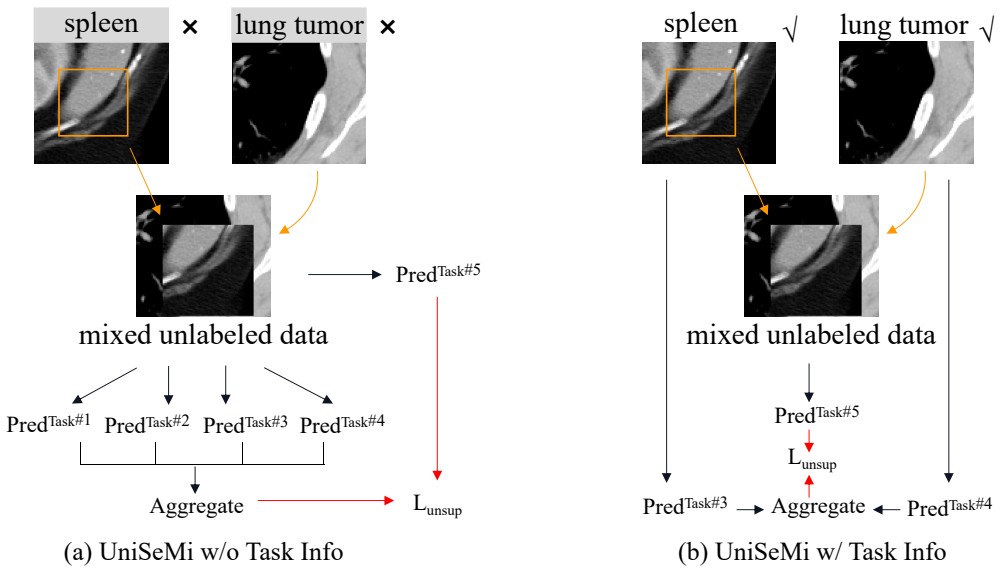

Figure 1: Workflow of UniSeMi w and w/o task infomation.

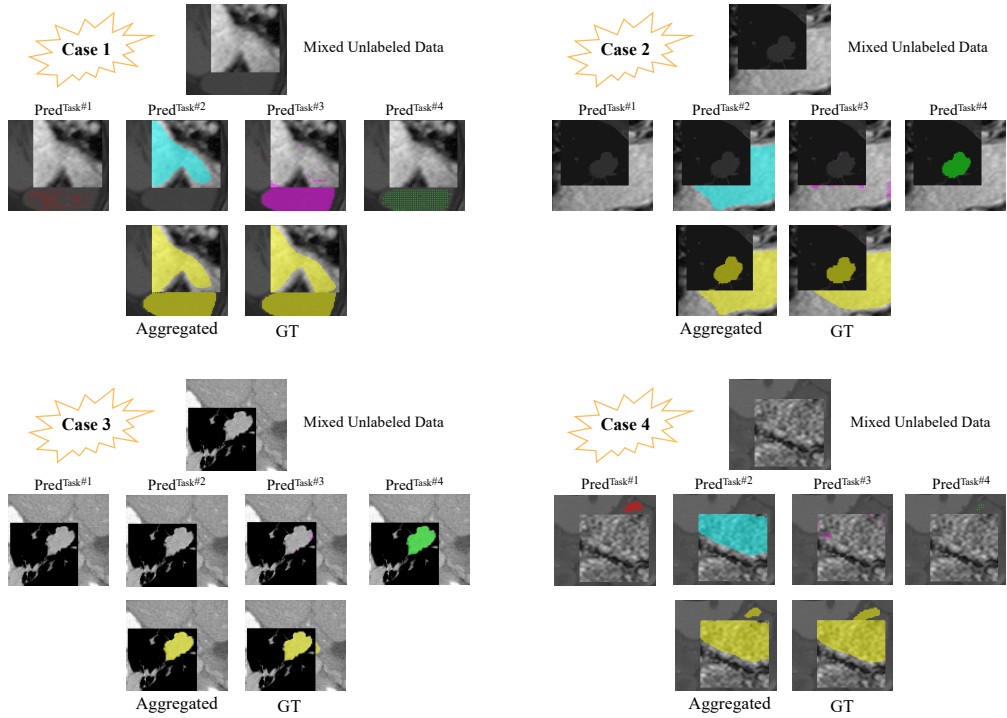

Figure 2: segmentation results prompted by pertinent tasks (Task#1~Task#4). Four random cases are presented. Case 1: cutmix between left atrium and spleen. Case 2: cutmix between left atrium and lung tumor. Case 3: cutmix between pancreas and lung tumor. Case 2: cutmix between pancreas and left atrium. **Task#1 to Task#4 are pancreas segmentation, left atrium segmentation, spleen segmentation and lung tumor segmentation, respectively.**

## D    VISUALIZATION OF SEGMENTATION ON FOUR BENCHMARKS

Fig. 3 shows the segmentation results, it is clear to find that UniSeMi can generate the most accurate mask compared to competitors. For example, for spleen segmentation (Row 5-6), other SSL methods extensively predict the background as the foreground, whereas UniSeMi successfully distinguishes the region of spleen. As for left atrium segmentation (Row 2-3), most SSL methods exhibit perfect results since the size of this dataset is larger than the others, thus models gain enough supervision from labels and smoothly generalize to test set.

## E    GENERALIZING TO UNSEEN TEST SET

We also conduct a experiment to assess the generalization ability of different SSL models. As Table 1 shows, BTCV-spleen is selected as the unseen test set, which contains 30 samples. Trained models (using 10% labels on the training set) are directly used to test and without any fine-tuning. Original results on MSD-spleen (Task#3) is marked by gray background, and the other side is the results on unseen test set BTCV-spleen. We can find UniSeMi consistently outperforms competitors on all metrics on the unseen test set. And there are approximately 2% performance degradation on Dice score for other methods, when compared the results on BTCV-spleen to MSD-spleen. Whereas no drop is found for UniSeMi. This phenomenon explicates UniSeMi can handle data from unknown sources more effectively, demonstrating the necessity of learning a unified SSL model.

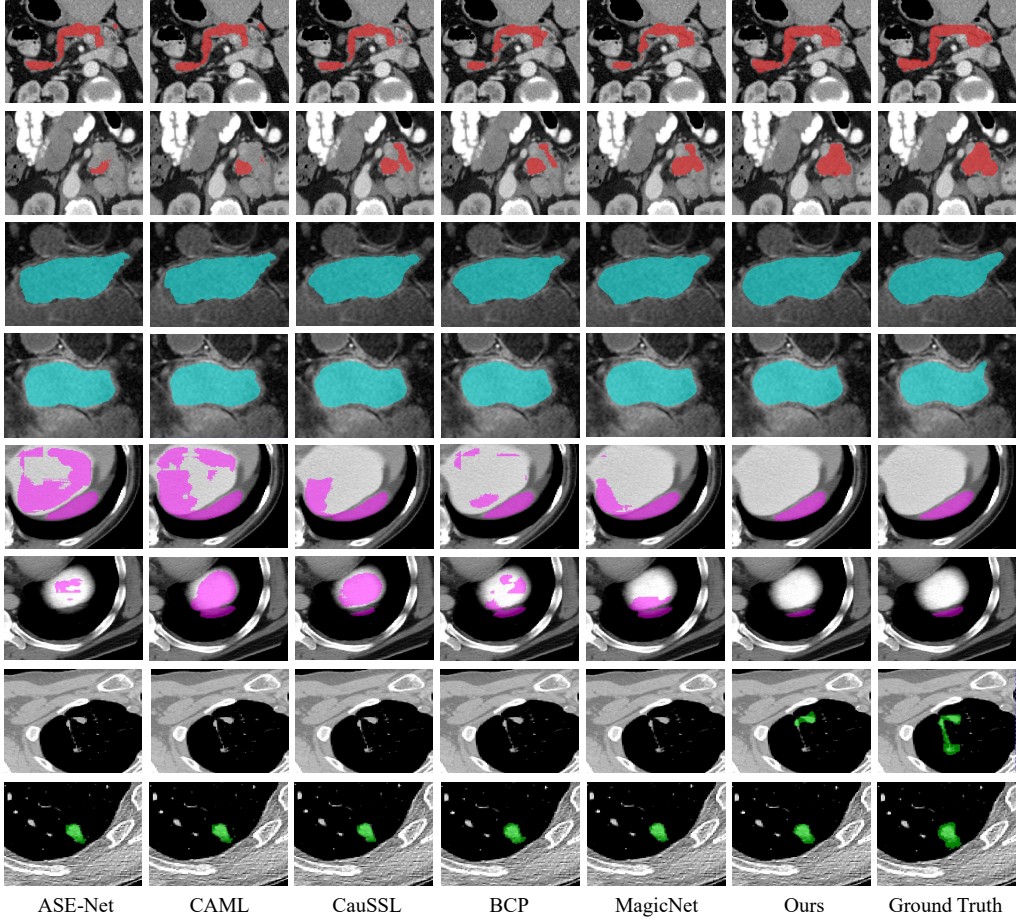

| ASE-Net | CAML | CauSSL | BCP | MagicNet | Ours | Ground Truth |

Figure 3: Segmentation results produced by different methods. Row 1-2: pancreas segmentation; Row 3-4: left atrium segmentation; Row 5-6: spleen segmentation and Row 7-8: lung tumor segmentation.

Table 1: A case experiment to see the model performance when facing unseen test set. Note that results are reported using the trained model with 10% labeled data on training set, no fine-tuning is conducted. BTCV-spleen and MSD-spleen are unseen test set and training set, respectively.

| Method | Directly testing on BTCV-spleen | | | | Original results on MSD-spleen (Task#3) | | | |
|---|---|---|---|---|---|---|---|---|
| | Dice ↑ | Jaccard ↑ | ASD↓ | 95HD ↓ | Dice ↑ | Jaccard ↑ | ASD↓ | 95HD ↓ |
| VNet | 73.94 | 61.12 | 14.48 | 43.82 | 75.14 | 65.27 | 15.02 | 43.89 |
| MagicNet (CVPR'23) | 80.94 | 70.15 | 14.98 | 45.08 | 83.55 | 73.58 | 13.49 | 41.79 |
| CauSSL (ICCV'23) | 79.34 | 67.78 | 16.89 | 48.02 | 81.98 | 71.25 | 14.69 | 41.84 |
| BCP (CVPR'23) | 81.87 | 71.52 | 14.39 | 44.22 | 83.12 | 72.85 | 14.42 | 42.11 |
| UniSeMi (Ours) | **89.36** | **83.01** | **2.91** | **10.93** | 89.34 | 81.73 | 3.12 | 9.33 |