# OpenReview forum: "UniSeMi: Toward Unified Semi-supervised Medical Image Segmentation"
_ICLR.cc/2024/Conference — ICLR 2024 Conference Withdrawn Submission_

### Official Review · Reviewer_wxyu · 2023-10-29

**Soundness:** 2 fair
**Presentation:** 3 good
**Contribution:** 2 fair
**Rating:** 5
**Confidence:** 4

**Summary:**

The paper proposes a unified semi-supervised learning (SSL) model called UniSeMi for medical image segmentation. The motivation behind UniSeMi is to address the limited availability of annotated data by leveraging both labeled and unlabeled data in a unified framework. It achieves this by augmenting the label space and using a task-prompted dynamic head. Additionally, UniSeMi can learn from unlabeled data without requiring task-specific information. The authors synthesize an additional task using labeled data from relevant tasks, which helps UniSeMi capture task semantics more effectively. The proposed UniSeMi model is evaluated on four public medical benchmarks, and the experimental results demonstrate its superiority over the second-best SSL method in terms of Dice and HD scores.

**Strengths:**

- Good Motivation: The idea of proposed UniSeMi model is to handle multiple tasks using a single model with a task-prompted dynamic head. This simplifies the training process and reduces the complexity of managing multiple models for different tasks.
- Performance Improvement: The experimental results demonstrate that UniSeMi outperforms the second-best SSL method in terms of Dice and HD scores on four public medical benchmarks.

**Weaknesses:**

- Limited Novelty: This work is essentially a semi-supervised learning adaptation of Dodnet [1], as it shares the same dynamic convolution approach with task prompt.
- Task Differentiation: Although there are four different segmentation tasks in the paper, they primarily differ in terms of organ type. It would be beneficial to treat them as a single task and expand the scope of UniSeMi to include other tasks such as registration and detection.
- Data Split and Model Stop: Section 4.1 only addresses the division of data into training and testing sets, leaving out the consideration of a validation set. So how to determinate the model stop criteria?
- Limited Dataset Size: The dataset used in the paper consists of only around 300 CT scans. To ensure the scalability and applicability of the proposed pipeline, it is crucial to explore larger CT datasets, such as AMOS, WORD, and AbdomenAtlas-8K, which offer a substantial number of CT scans.
[1] Dodnet: Learning to segment multi- organ and tumors from multiple partially labeled datasets

**Questions:**

See weakness part.

---

### Official Review · Reviewer_6ofX · 2023-10-31

**Soundness:** 2 fair
**Presentation:** 3 good
**Contribution:** 2 fair
**Rating:** 3
**Confidence:** 5

**Summary:**

This paper proposed a unified framework that can (1) handle different segmentation tasks using task prompt (i.e., to segment different classes shipped with datasets) and (2) train a single model using both labeled and unlabeled data. First, the task prompt follows the similar technique introduced by DoDNet (Zhang et al., 2021). Second, the semi-supervised learning follows the similar strategy as CutMix (Yun et al., 2019), but integrated with different task prompts using a consistency loss. The technical contribution is to add semi-supervised strategies to DoDNet, enabling the model to effectively leverage both labeled and unlabeled data. Results on four public datasets show improved performance over previous semi-supervised methods.

**Strengths:**

+ The method is clearly described and appropriate citations are provided when techniques are derived from prior research.
+ The concept of the method is sound, effectively ensuring consistency across disparate task prompts through the integration of CutMix.
+ The comparison with previous semi-supervised methods appears comprehensive.
+ The experimental setting (e.g., dataset split) is clearly described, suggesting that replication of the results would be straightforward, especially with the forthcoming release of the source code.

**Weaknesses:**

- The use of one-hot vector for task prompt appears to be suboptimal (see details in Q1).
- Lack of baseline methods that sharing similar technique to task prompt (see details in Q2).
- The implementation of the proposed method is unclear (see details in Q3).
- The comparison with previous semi-supervised methods seems unfair (see details in Q4).

**Questions:**

1. The task is defined as dataset in this paper (Section 4.1), but in practice, there are overlapping classes between datasets. E.g., both NIH-Pancreas and MSD-Pancreas datasets have pancreas segmentation task. Also, there are difference across datasets. E.g., MSD-Pancreas requires pancreatic tumor segmentation, but NIH-Pancreas does not. With this, defining task using one-hot vector seems suboptimal. Can authors comment on this issue?
2. The effectiveness of task prompt, as one of the technical components (contributions), has not been verified. The authors should compare the task prompt with previous approaches that can also unify multiple datasets/classes. E.g., how does it differ from DoDNet, CLIP-Driven Universal Model, MultiTalent, UniSeg, and many more?
3. In Tables 1-4, the term "w/ Task Info" is somewhat ambiguous. What kind of information—is it the one-hot vector? Also, how do you implement UniSeMi (without Task Info)? Without making these settings clear, it is impossible to assess the value of the proposed method.
4. In Tables 1-4, semi-supervised methods are only trained on a single dataset, but UniSeMi is trained on a combination of four datasets. The comparison is unfair and the performance improvement is marginal and even worse (e.g., Table 2).

---

### Official Review · Reviewer_jVTd · 2023-11-01

**Soundness:** 2 fair
**Presentation:** 2 fair
**Contribution:** 3 good
**Rating:** 5
**Confidence:** 4

**Summary:**

This paper tackles a problem in task-specific supervised image segmentation and presents a unified semi-supervised segmentation model that segments various regions via a dynamic head with task-specific prompts. Also, by leveraging labeled data, the authors generate synthetic data that can be used for training the model to segment semantic regions. This enables the model to achieve unlabeled data learning. The proposed method is demonstrated on four public datasets and shows relatively superior performance compared to the existing comparative methods.

**Strengths:**

- The paper proposes a model that can perform various image segmentation tasks such as spleen, pancreas, and lung tumor segmentation, using task-specific prompts.
- the proposed method can leverage unlabeled data by utilizing labeled training data in learning semantic segmentation.
- The proposed method is compared to the recent segmentation methods and achieves superior performance.
- The proposed UniSeMi is analyzed by various studies of copy-past choices, distribution on labeled and unlabeled data, and simultaneous task-specific segmentation from other SSL methods.

**Weaknesses:**

- It is difficult to figure out how to construct a synthetic task. If combining two images from two different tasks in equation (3), the synthetic image can have discontinuities in the boundaries of the mask region. Please provide several examples of synthetic images and labels.
- In Section 3.3, the task#5 is not described in detail. Also, if the images in task#5 are different from the images in task#1-4, the prediction of task#5 and the aggregated prediction of task#1-4 would be different. How is the consistency applied between those two predictions?
- The experimental results of Table 1 and Table 2 only use a small number of training data. How is the result different according to the amount of training data?
- There are many methods [1,2,3] that can use unlabeled data without pseudo masks, but they are not compared.

[1] Bortsova, Gerda, et al. "Semi-supervised medical image segmentation via learning consistency under transformations." Medical Image Computing and Computer Assisted Intervention–MICCAI 2019: 22nd International Conference, Shenzhen, China, October 13–17, 2019, Proceedings, Part VI 22. Springer International Publishing, 2019.

[2] Kim, Boah, and Jong Chul Ye. "Mumford–Shah loss functional for image segmentation with deep learning." IEEE Transactions on Image Processing 29 (2019): 1856-1866.

[3] Zhu, Lei, et al. "Semi-supervised unpaired multi-modal learning for label-efficient medical image segmentation." Medical Image Computing and Computer Assisted Intervention–MICCAI 2021: 24th International Conference, Strasbourg, France, September 27–October 1, 2021, Proceedings, Part II 24. Springer International Publishing, 2021.


*Minor:*

It would be better if each table and figure could be placed in the right position as much as possible. For example, Table 7 can be placed in the paragraph of Copy-past choices of Task#5.

**Questions:**

Please see the above weaknesses.

---

### Official Review · Reviewer_Nu9m · 2023-11-03

**Soundness:** 2 fair
**Presentation:** 2 fair
**Contribution:** 2 fair
**Rating:** 5
**Confidence:** 4

**Summary:**

The paper introduces an innovative approach that integrates multiple datasets to learn from both supervised and synthetically supervised signals, aiming to enhance segmentation performance significantly over existing methods. The core of the proposed methodology is a dynamic convolution mechanism informed by task-specific prompts, which tailors multi-head architectures to their respective tasks (sec. 3.1). To enhance the learning, the authors construct a synthetic dataset and implement a supervised loss, which is pivotal for advancing Task #5 (sec. 3.2). This, in turn, supports more effective learning from unlabeled data, utilizing techniques like CutMix (sec. 3.3). The authors assert that their method has achieved notable results across four distinct segmentation tasks.

**Strengths:**

The paper demonstrates impressive experimental outcomes, showcasing an innovative aspect in its approach.

Originality: The creation of a new supervised learning task (Task #5) stands out as a noteworthy contribution, facilitating the utilization of unlabeled images to improve the performance of tasks #1-4. This concept exhibits the potential for broad applicability across various domains within this paradigm, marking a creative angle in addressing semi-supervised learning challenges.

Quality: The research is underpinned by a robust experimental framework, contrasting the proposed method with a diverse array of existing approaches, including several recent and strong baselines. The experimental evidence provided appears to solidly back the authors' claims, demonstrating the method's effectiveness.

Clarity: While the paper is well-structured, the presentation of the central idea lacks clarity and could benefit from additional detail. I encountered difficulty in fully grasping the nuances of the proposed approach, a challenge not mitigated by the supplementary materials, which do not offer the necessary elucidation.

Significance: The paper's outcomes are indeed promising; however, I ascribe a moderate level of significance to this work. This stems from the lack of method's details and the justification for the experimental setups. A more thorough explanation of the choices made in the experimental design would better substantiate the reported results and their implications for the field.

**Weaknesses:**

Presentation and Detailing: The paper's innovative idea requires more rigorous exposition to be fully understood. The current presentation lacks the requisite detail to grasp the intricacies of the method. I will raise the points in the following section.

Literature Review: The paper's discussion of related work appears to be short, engaging with a limited selection of recent studies. A more comprehensive review of the literature, including seminal works and cutting-edge research, would contextualize the contributions of this paper within the broader field.

Experimental Design and Validation: The methodology for experimental validation requires further elaboration. Specifically, there is an absence of a detailed justification for the chosen hyperparameters and the data split for training and testing phases. This leaves open questions about the potential for overfitting and the generalizability of the results.

**Questions:**

I will be happy to raise my evaluation if the authors can answer my following questions:

- The universal models are categorized into “task-agnostic” and “task-specific”, to which category the UniSemi framework belongs?

- what is \phi_{\theta} in Equ. 1, there is no such variable in Equ. 1.

- the dynamic convolution seems not proposed by the authors, where does it come from? One should strictly cite the original paper.

- What is the network used in this work? I see extensive experiments carried out on V-Net, does it imply that the authors used V-Net as the backbone? Where is the dynamic convolution in this architecture, and does it have any hyperparameter to tune?

- Could the authors confirm whether the five tasks are trained jointly for Equ. 5?

- I see that the creation of synthetic dataset (#task 5) requires data from different tasks to be mixed up. Does this procedure require any registration of images? Why and how this can impact the experimental results?

- Can the authors explain why they just split the data into two sets and how did they manage to report unbiased results?

- HD metric is usually reported in absolute distance, such as mm or pixels. I am wondering why HD is in percentage in this work.